# Novel Tetraploid Triticale (Einkorn Wheat × Rye)—A Source of Stem Rust Resistance

**DOI:** 10.3390/plants12020278

**Published:** 2023-01-07

**Authors:** Michał T. Kwiatek, Aleksandra Noweiska, Roksana Bobrowska, Adrianna Czapiewska, Mert Aygün, Francois d’Assise Munyamahoro, Sylwia Mikołajczyk, Agnieszka Tomkowiak, Danuta Kurasiak-Popowska, Paweł Poślednik

**Affiliations:** 1Department of Genetics and Plant Breeding, Poznań University of Life Sciences, Dojazd 11, 60-632 Poznań, Poland; 2Institute of Plant Genetics, Polish Academy of Sciences, Strzeszyńska 34, 60-479 Poznań, Poland; 3Agricultural Research Station, Poznań University of Life Sciences, Dłoń 4, 63-912 Miejska Górka, Poland

**Keywords:** einkorn, fluorescence in situ hybridization, genetic diversity, resistance genes, steam rust, triticale

## Abstract

Among cereals, triticale (×*Trititcoseale* Wittmack ex A. Camus) represents a number of advantages such as high grain yield even in marginal environments, tolerance to drought, cold and acid soils, as well as lower production costs. Together with high biomass of grain and straw, triticale is also considered as an industrial energy crop. As an artificial hybrid, it has not evolved naturally, which is reflected in narrow genetic diversity causing a resistance collapse in recent years. Here, we describe a novel, synthetic tetraploid triticale, which was developed by the crossing of rye (*Secale cereale* L.) with einkorn wheat (*Triticum monococcum* spp. *monococcum*), which possess *Sr35* stem rust resistance gene. Three subsequent generations of alloploids were obtained by chromosome doubling followed by self-pollination. The cytogenetic analyses revealed that the amphiploids possess a set of 28 chromosomes (14 of Am-genome and 14 of R-genome). The values of the most important yield-shaping traits for these tetraploid triticale form, including thousand-grain weight, plant height and stem length were higher compared to parental genotypes, as well as standard hexaploid triticale cultivars. This study shows that this tetraploid triticale genetic stock can be an interesting pre-breeding germplasm for triticale improvement or can be developed as a new alternative crop.

## 1. Introduction

Modern agriculture is characterized by a significant genetic base erosion. Thus, there is a need to access a wider range of crops with the extended genetic base for functional traits improvement, which meets the global market requirements, considering the current demand for renewable energy sources and increasing prices of agricultural inputs (e.g., fertilizers, chemicals, energy, etc.). Triticale (×*Trititcoseale* Wittmack ex A. Camus) is a cultivated allopolyploid, which was obtained through hybridization of wheat derivatives (*Triticum* sp.) with diploid rye (*Secale cereale* L.). The interest in triticale production has increased because of its potential to become an industrial energy crop [1]. The economic importance of triticale is reflected by a significant acreage in Europe (3,392,695 ha) which constitutes ca. 90% of the world production (3,812,724 ha) [2]. World energy demand is a major issue for science in the 21st century (UNIDO, Advisory group on energy and climate change, 2010); hence, the use of triticale biomass as feed-stock for anaerobic digestion provides innovative solutions, considering energy needs. As a non-food crop, it could substitute current uses of corn and wheat for ethanol and co-product production [3]. A more extensive and deeper rooting system, which is characteristic to rye, makes triticale potentially better than either barley or wheat, especially on less fertile, poor soils. However, as an artificial crop, triticale is characterized by the low genetic diversity. Moreover, increasing the harvesting area of this crop is associated with the rapid development of fungal pathogens which are continuously adapting to triticale. Among fungal pathogens, stem rust of wheat caused by *Puccinia graminis* f. sp. *tritici* (*Pgt*) seems to be one of the most devastating fungal diseases. Since aggressive new strains have emerged, such as *Ug99*, first detected in Uganda in 1999 [4], this pathogen has revealed the rapid expanse with the potential to result in a 100% yield loss on susceptible varieties. *Pgt* appeared on triticale soon after commercial cultivation started [5]. Resistance in triticale to stem rust has been reported and several resistance genes of rye origin such as *Sr27*, *Sr31*, *Sr1R^Amigo^*, *Sr50 SrNin*, *SrSatu*, *SrJ*, *SrBj* and *SrVen* have been described [6]. First, four genes were already transferred to wheat. What is more, the close genetic relationship of wheat and rye is not only restricted to the host side of the pathosystem [7]. Two forms (*formae speciales*) of *P. graminis*, specific to rye (*secalis*) and wheat (*tritici)* can hybridize and probably also had a common ancestor [8].

So far, little is known about the infection process of triticale by *Pgt.* Recent studies have shown that triticale cultivars carrying *Sr27*, *SrKw* and *SrSatu* remained resistant to *Pgt* pathotypes in the *Ug99* lineage [9,10]. The *Pgt* race *Ug99*, later characterized as race TTKSK [11] exhibits unique virulence patterns and possesses virulence to a wide range of resistance genes of both wheat and wheat-related origin [12]. The virulence combination in race TTKSK rendered more than 90% of global wheat varieties susceptible during testing in 2005 [13]. Moreover, multiple derivatives of TTKSK have been reported, including: TTKST, virulent to *Sr24* and identified in 2006 [14]; TTTSK, virulent to *Sr36* and identified in 2007; and TTKTT, virulent to *SrTmp* and identified in 2014 [15]. The appearance of Ug99 race group clearly highlights the boom-and-bust cycle by the defeat of important R-genes, such as *Sr24*, *Sr31*, *Sr36*, and *SrTmp*, which were widely deployed worldwide [15]. Hence, new resistance gene combinations need to be assembled.

New plant breeding techniques (NPBTs), which are based on gene or genome editing, have evolved rapidly in recent years, providing much faster protocols and more precise results than conventional plant-breeding techniques. However, in the European Union, there is considerable debate as to how these new techniques should be regulated, and whether some or all of them should fall within the scope of legislation on genetically modified organisms (GMOs). Therefore, non-GMO approaches in crop improvement programs are still widely used. The exploration of gene pools of species related to cultivated plants plays a crucial role in crop improvement. Distant hybridization can break species limits, increase genetic variation, and combine the biological characteristics of existing species.

Einkorn wheat (*Triticum monococcum* L.; 2n = 2x = 14 chromosomes, A^m^A^m^) is considered as a primary gene pool of cultivated wheats and can refer either to the wild species of wheat, *Triticum boeoticum*, or to the domesticated form, *Triticum monococcum*. Cultivated einkorn is represented by a board genetic variation [16]. This taxon includes nearly twenty identified botanical varieties and six eco-geographical groups [17]. Geographical diffusion of einkorn from the region of domestication is well reconstituted thanks to grain remains found in archaeological excavations and is an example of well documented speciation on the background of time and location [18]. This diploid species is closely related to *Triticum urartu* Thumanjan ex Gandilyan (2n = 2x = 14; A^u^A^u^), which is reported as a one of the ancestors of common wheat (*Triticum aestivum* L.; 2n = 6x = 42; AABBDD). Cultivated *T. monococcum* L. subsp. *monococcum* originated from *T. boeticum* (syn. *T. monococcum* subsp. *aegilopoides*), which was widely distributed in southern Europe and western Asia [17]. Its common name comes from the German “Einkorn”, which means ‘single grain’, and refers to the presence of only one grain per spikelet [16]. This cereal was important in the Neolithic early agriculture, but is now extensively grown in western Turkey, the Balkans, Switzerland, Germany, Spain, and the Caucasus [19]. During last 5000 years, einkorn was discarded and replaced by tetra- and hexaploid wheats. Currently, einkorn wheat is widely used to improve wheat genetic variability, among others to transfer the rust resistance genes, including stem rust resistance gene *Sr21*, *Sr22*, *Sr35 and Sr60* [20].

In early 1980s, the potential of diploid wheats for triticale improvement was noticed. Neumann [21] proposed three approaches of diploid wheat chromatin introduction into triticale genetic background: (1) direct crossing of hexaploid triticale with diploid wheat; (2) crossing of tetraploid wheat with diploid wheat, followed by crossing with hexaploid triticale; (3) direct crossing of diploid wheat with diploid rye, followed by crossing with hexaploid triticale. Sodkiewicz [22] used the artificially developed *T. monoccocum* × *S. cereale* amphiploid as a pollen donor for crossing with hexaploid triticale. It was the first successful attempt to transfer the A^m^ genome of *T. monococum* (line Tm16) into triticale background to increase the leaf rust resistance [23]. Furthermore, this germplasm (A^m^A^m^RR) was used for the development of secondary tetraploid genotypes of triticale carrying different substitutions of A^m^ by A chromosomes, which was introduced into hexaploid triticale to transfer leaf rust resistance genes [22].

The aim of this study was to create an amphidiploid form of triticale (eikorn wheat × rye; 2n = 4x = 28 chromosomes, A^m^A^m^RR) carrying the effective *Sr35* resistance stem rust gene against the *Ug99* race, which can be used as a prebreeding germplasm for triticale improvement. In this work, we present: (a) the identification of *NL9F5* molecular marker linked to *Sr35* loci in the collection of *T. monococcum* accessions; (b) the development of amphidiploid forms through distant cross-hybridization followed by self-pollinations, (c) karyotyping of parental forms, hybrids and amphidiplioids; (d) the molecular identification *Sr* resistance genes in einkorn × rye amphiploid form (A^m^A^m^RR); (e) the analysis of chromosome dynamics during meiosis of pollen mother cells of amphiploid forms; (f) the evaluation of stem rust resistance level of amphiploid forms; and (g) the analysis of key yield-shaping traits.

## 2. Results

### 2.1. Identification of Molecular Markers Linked to Sr22 and Sr35 loci

The PCR analyses were performed for 24 accessions of *T. monococcum* in two replications, and the results of both trials were identical (Table 1). The *Sr35* stem rust resistance gene was identified only in one accession (PI 191383) using *NL9F5* marker. The PI 191383 accession amplified a 719 bp band which is linked to carrying *Sr35* resistance allele (Figure 1). As regards remaining 23 accessions, the PCR products had the same size equal to ca. 800 bp.

### 2.2. Cross-Hybridizations and Self-Pollinations

Six cultivars of winter rye were used as a pollen donors for the cross-hybridization procedure with the selected einkorn wheat accession (PI 191383), which carried *Sr35* stem rust resistance gene. Pollen of each rye cultivar was used to pollinate twenty spikes of chosen einkorn accession. In total, 5088 florets (120 spikes) of einkorn wheat were pollinated, and 243 hybrid seeds were obtained (Table 2). The total crossing efficiency was ca. 5%. More specifically, the range of pollinated einkorn florets considering the particular pollen donor (rye cultivar) ranged between 766 and 916. The number of immature F_1_ embryos ranged between 36 and 47, considering specific crossing combinations (Table 2). A total of 243 F_1_ embryos were suited for regeneration in in vitro conditions; however, only 163 developed the roots, of which 141 plants matured and were treated with colchicine solution. Spikes of F_1_ plants were subjected to a cytogenetic evaluation. The majority of florets were infertile. Considering the molecular marker analysis, only one crossing combination, PI 191383 × Hadron, was later developed. The spikes of S_2_ plants were isolated using paper bags to secure self-pollination. A total amount of 79 S_3_ seeds were harvested. Thereafter, S_3_ seeds were germinated and 68 mature plants were developed (Table 3). Chromosome pairing during meiosis of pollen mother cells was examined.

### 2.3. Karyotyping

At first, karyotypes of *T. monococcum* PI 191383 accession (Figure 2a) and rye cv. Hadron (Figure 2b) were developed in order to track and identify all chromosomes in *T. monococcum × S. cereale* hybrids. The probe signals of pTa-86 were identified in the telomeric region of long arms of 2A^m^ and 4A^m^ chromosomes, and were abundant across the rye chromosomes. As opposed to pTa-86 signals, pTa-535 probe landmarks were observed in all *T. monococcum* chromosomes and were most informative for the analysis of A^m^-genome. However, none of pTa-535 bands were observed on R-genome chromosomes. The less abundant signals came from pTa-713 probe, and were located in 4A^m^L and 6A^m^ L (telomeric region), 3A^m^ and 7A^m^ (pericentromeric region) and 6RS.

In all S_1_ hybrids, the number of chromosomes was 14. GISH experiments confirmed that each of two subgenomes is represented by a haploid number of chromosomes (7 chromosomes of A^m^-genome and 7 chromosomes of R-genome) (Figure 3a). Thereafter, only two plants of S_2_ generation were germinated and developed, and hence, it was only possible to analyse those two plants. The total number of chromosomes was 28 (14 chromosomes of A^m^-genome and 14 chromosomes of R-genome; Figure 3b). Moreover, the FISH analysis showed that each homeologous group is represented by two copies of chromosomes. Similar chromosome set was characteristic to all 68 S_3_ plants.

### 2.4. Meiosis Analysis in Pollen Mother Cells

The examination of homologous chromosome pairing was performed for two plants of S_2_ generation and 10 plants of S_3_ and S_4_ generation of PI 191383 x Hadron crossing combination (Table 4; Figure 4). The range of 0–2 multivalents involving *S. cereale* chromosomes was observed in pollen mother cells of S_2_ plants during metaphase I (Figure 4a). There was no rye univalent observed. From the other side, *T. monococcum* chromosomes did not form multivalents in S_2_ plants; however, univalents were observed. The process of chromosome pairing in pollen mother cells of S_3_ plants was more stable, comparing to prior generation. In most cases, 7 bivalents of rye chromosomes and 7 bivalents of *T. monococcum* chromosomes were observed (Figure 4b). However, einkorn chromosome univalents and rye chromosome multivalents were sporadically detected (Table 4).

### 2.5. Ptg Infection Tests

Sixty plants of S_4_ generation were divided into three experimental trials (twenty S_4_ plants per trial). Each trial was enriched with following control genotypes: *T. aestivum* “Thatcher”, (susceptible control, 10 plants) and *T. aestivum* Thatcher+*Sr57* (moderate resistant control, 10 plants) (Figure 5). The range of infection scores was 0–1 for S_4_ plants belonging to all three repetitions. There were no significant differences between the repetition’ means (F-ratio = 0.12; *p* = 0.89), and all plants possessed the resistant allele for *Sr35* gene (Table 5).

In contrast, the wheat cv. “Thatcher” were highly infected which reflects in the infection score, which ranged between 3 and 4. No significant differences between group means was detected (F-ratio = 0.26; *p* = 0.77). The moderate resistance against *Ptg* was observed within Thatcher + *Sr57* plants in all groups. In this case, the infection scores ranged between 1 and 2 (Table 5) and there were no significant differences between group means noticed. Significant differences for mean scores were observed between the amphiploids and control genotypes (Appendix A).

### 2.6. Evaluation of Yield Related Traits in Progeny of S_3_ Generation Plants

Statistics of eight yield-related traits were investigated among S_4_ plants (Table 6, Figure 6 and Figure 7). Confidence intervals (CI) for estimated mean of population were calculated at 0.95 and 0.99 significance levels. The two-tailed critical value was equal to 1.99 for the α = 0.05 level of significance in the case of a 0.95 CI; and 2.63 for the α = 0.01 level of significance in the case of a 0.99 CI. Considering the biomass characterization, plant height (PHt), stem length (StL) and thousand-grain weight (TGW) are the most important factors. The mean plant height was ca. 154.1 cm. It was higher than the mean plant height for both parental forms (einkorn wheat, 72.53 cm; rye, 152.78 cm). Despite the wide range of amphiploids height scores (149–162 cm), the standard deviation and standard error were 3.5006 and 0.4519, respectively, which were reflected in the relatively narrow confidence intervals for estimated mean of the population 152.1 ± 1.6 cm at α = 0.95 and 152.6 ± 2.1 cm at α = 0.95. The mean scores for stem length (StL) showed the same manner, the value of this parameter for amphiploids was higher compared to parental forms. The most important yield parameter, thousand-grain weight (TGW), varied between 25.1 and 70.2 for amphiploidy plants (Table 6). The TGW mean for tetraploid triticale was 39.5 g, which was higher than mean TGW for parental components: Dankowskie Hadron rye (35.8 g) and einkorn wheat (30.2 g).

## 3. Discussion

In classical breeding, crossing barriers are considered as limiting factors for the effective transfer of genes between species differing in terms of ploidy level. The attempts of direct hybridization between diploid *T. monococcum* (2n = 14 chromosomes, A^m^A^m^) and hexaploid triticale (2n = 42, AABBRR) failed [21]. To overcome those obstacles, diploid forms are crossed with *Triticum turgidum* (2n = 28, AABB) followed by cross-hybridization of triploids (A^m^AB) or doubled haploids (A^m^A^m^AABB) with triticale [20]. In this study we reported the direct cross-hybridization of *T. monococcum* (2n = 14, A^m^A^m^) with diploid rye (2n = 14, RR). This pathway is considered as the most effective way to improve many polygenic traits [24]. Amphidiploids can serve as bridges for introgression alien genes and are developed by the union of unreduced gametes [25]. The formation of unreduced gametes is the most important mechanism for chromosome doubling in distant hybrids and the main driving force in generating polyploidy species in nature [26]. This mechanism can be induced artificially by colchicine treatment [27]. A number of attempts for crossing diploid wheat with diploid rye were performed without any success [21]. However, Sodkiewicz [22] succeed in generation of A^m^A^m^RR amphiploid. For this purpose, diploid wheats (*T*. *monococcum* and *T. boeticum*) were used for crossing with rye cv. “Dańkowskie Złote”. By application of various modifications of cross-hybridization techniques (gibberellic acid spraying, different pollination times) first artificial tetraploid triticale (A^m^A^m^RR) by crossing T. *monococcum* L. var *macedonicum* Pagag. with diploid rye followed by chromosome doubling using colchicine treatment. In the present study, the crossing efficiency was much lower (CE = 0.04–0.05, Table 2) compared to the results of Sodkiewicz [22] (CE = 3.23–19.15), but the embryo-rescue technique allowed us to regenerate 141 plants (Table 3).

The karyotype analysis using repetitive sequence probes was significant to evaluate the chromosome composition of haploid generation and S_1_–S_3_ generation plants. The following probes used in the study, namely, pTa-86, pTa-535, pTa713, were previously successfully utilized for the chromosome identification of *T. monococcum* [28], wheat [29] and triticale [30,31,32,33]. What is more, genomic in situ hybridization with the genomic probes enables us to conclude that the subsequent generations (S_2_ and S_3_) of amphidiploids are more cytologically stable than S_1_ generation plants. The chromosome pairing disorders, chromosome imbalances and genome instability can result because of induced allopolyploidisation, which can lead to “genome shock” [34] of the newly formed amphidiploids, which causes extensive genetic and epigenetic changes in the nascent hybrids [35]. It is reported that genome stabilization in further generations can be provided by non-random elimination of genes and noncoding sequences, which induced differentiation and diversity of different genomes or chromosomes in subsequent generations, which is required for successful homology recognition and meiotic pairing [36].

In this study, we have obtained a cytogenetically stable amphiploid form which carries a *Sr35* effective stem rust resistance gene, derived from selected *T. monococcum* PI 191383 (Table 1). The PI 191383 accession originates from Ethiopia, the region where many aggressive races have been detected, including TTKSK (1st detection in 2003), as well as TTTSK (2007), PTKSK (2007) and PTKST (2007) [13]. This accession was previously screened together with other 1060 accessions of *T. monococcum* against race TTKSK (*Ug99*) and four additional *P. graminis* f. sp. *tritici* races [37]. It was reported that PI 191383 is resistant against TTKSK (Ug99), as well as TRTTF, QFCSC, TTTTF and MCCFC, and it is postulated to carry *Sr21* and *Sr35* loci. So far, Singh et al. [11] reported that *Sr21* is no more an effective gene against *Ug99* lineage (race TTKSK), while *Sr22* and *Sr35* are still effective, but the virulence for the latter one gene is known to occur in other races. Here, we confirmed the presence of *Sr35* gene using molecular markers in PI 191383 accession and subsequent hybrid generations (haploids and amphiploids). The infection tests with natural inoculum, which was collected in Poland, revealed the significant difference in the reaction level between completely resistant A^m^A^m^RR amphiploids (S_4_ generation, *Sr22+Sr35*) compared to the following controls: *T. aestivum* “Thatcher + *Sr57*” (reported to be moderately resistant to stem rust, *Sr57*) and *T. aestivum* “Thatcher” (susceptible to stem rust) [38].

Triticale straw and seeds constitute a potential raw material for biofuel production. Moreover, at present, growing interest in organic agriculture practices is initiate the testing of triticale applicable to organic or low-input farming. In particular, triticale shows a number of advantages such as high grain yield even in marginal environments [39]. Hence, high yield and biotic stress resistance is necessary to meet the challenges of the market. Here, eight yield-shaping traits have been scored for S_4_ generation of amphiploids (A^m^A^m^RR) and compared to parental forms (einkorn and rye) (Table 5). The plant height and stem length means of amphiploids were significantly higher compared to parental accessions, while The same dependency was observed for stem length thousand-grain weight; these are the crucial factors considering the biomass evaluation. The thousand-grain weight (TGW) mean for tetraploid triticale was 39.5 g, which was higher than mean TGW for parental components: Dankowskie Hadron rye (35.8 g) and einkorn wheat (30.2 g). What is worth underlining is that the average TGW for commercial varieties of hexaploid triticale ranges between 39 g and 43 g [39].

## 4. Methods

### 4.1. Plant Material

Seeds of nineteen accessions of *Triticum monococcum* L. subsp. *monococcum* and five accessions of *T. monococcum* L. subsp. *aegilopoides* (Link) Thell. (National Small Grains Collection (USDA, Aberdeen, ID, USA) (Table 1), reported as sources of stem rust resistance genes, were used for distant cross-hybridizations with diploid rye (*S. cereale* L.).

Following six cultivars of winter rye: Piastowskie, Poznańskie, Antoninskie (Poznan Plant Breeding Sp. z o.o., Poznań, Poland), Hadron, Granat, and Turkus (Danko Plant Breeding Sp. z o.o., Choryń, Poland), were used as pollen donors. Seeds were germinated on Petri dishes. The plantlets were transferred into pots fulfilled with soil and were cultivated in short-day conditions (8 h light/16 h dark, 20°/18 °C). After 6 weeks, winter genotypes (rye accessions) were transferred to vernalizing conditions (10 h light/14 h dark, 4 °C) for 8 weeks and followed by long-day conditions (13 h light/11 h dark, 20°/16 °C). The F_1_, S_1_–S_3_ plant generations were grown in greenhouse at the Department of Genetics and Plant Breeding at Poznan University of Life Sciences (PULS), Poznań, Poland. The progeny of S_3_ plants were sown in five-row experimental plot (1 m^2^) in three replications at Agricultural Research Station of the PULS in Dłoń, Poland (51°41′29.0′′ N 17°04′32.9′′ E).

### 4.2. Cross-Hybridizations

Six rye (*S. cereale*) cultivars were used as pollinators. Forty seeds of each of rye cultivars were germinated in four runs every two weeks, in order to secure the sufficient amount of pollen and to catch the flowering time of *T. monococcum* genotypes. For cross-pollination, immature *T. monococcum* florets were emasculated using an embroidery scissors together with fine-point forceps and have been isolated with glassine crossing bags. When the female reproductive parts (pistils) of *T. monococcum* accessions were mature, the freshly extracted anthers of rye were used for pollination by entering each floret. Moreover, the spikes of pollen donor and acceptor were isolated together in one bag to increase the effectiveness of the cross-pollination process. After two days, the pollinated spikes were isolated using individual bags. Immature F_1_ embryos (10–14 days after pollination) were cultured on Gamborg’s B5 medium, as described by Ślusarkiewicz-Jarzina et al. [27]. The chromosomes of the F_1_ plants (dihaploids) were doubled through colchicine treatment [27] to produce amphidiploids S_1_. The S_1_ seeds were self-pollinated to generate fertile amphidiploids. Each generation (S_1_–S_3_) of hybrid plants was evaluated through cytological analysis.

### 4.3. Identification of NL9F5 Molecular Markers Linked to Sr35 Loci

Leaf tissues were harvested from young leaves of parental forms and F_1_ hybrids, and genomic DNA was extracted using Plant & Fungi DNA Purification Kit (EurX, Gdańsk, Poland). The quality and quantity of DNA was assessed using a NanoDrop 8000 spectrophotometer (Thermo Scientific, Waltham, MA, USA). The DNA concentrations were optimized up to 100 µg/µL. Total genomic DNAs were used as templates for PCR. The reaction was performed in 25 μL reaction mixture containing: 50 ng/μL of DNA, 2 × TaqNova HS buffer (4 mM MgCl_2_, 0.4 mM of each of dNTPs and 0.1 U/µL DNA TaqNovaHS polymerase) (Blirt, Poland), 12.5 pmol of each primer (Table 2), and MQ H_2_O, Amplifications were carried out in LabCycler thermocycler (SensoQuest Biomedizinische Elektronik, Goettingen, Germany) using the amplification temperatures presented in Table 2. Amplification products were electrophoresed at 5 V/cm for about 3 h in 1.5% agarose gel (Sigma), stained Midori Green Direct (Nippon Genetics Europe GmbH), visualized under UV light and photographed (Gel-Doc imagining system, Bio-Rad).

### 4.4. Preparation of Chromosome Spreads

Mitotic metaphase accumulation and fixation procedures were carried out according to Kwiatek et al. [40]. Briefly, the root tips of parental and hybrid plants were collected to Eppedorf tubes fulfilled with tap water when the roots reached 1–2 cm in length. Accumulation of metaphase chromosome was induced by ice-cold water incubation for 26 h followed by chromosome fixation, using ethanol:acetic acid (*v*/*v* 3:1) fixation solution and stored in the freezer (−20 °C).

Spikes of the S_1_ hybrids and S_3_ generation were fixed in Carnoy’s solution (ethanol: chloroform: acetic acid, 6:3:1) for 24 h and then stored in 70% ethanol. The anthers were squashed in 60% acetic acid for meiotic studies. Pollen mother cells (PMCs) at meiotic metaphase I were observed for chromosomal pairing. One hundred cells of each plant were observed. The observations were made and documented with a Delta L-1000 contrast-phase microscope (Delta Optical, Nowe Osiny, Poland) coupled with a DLT CamPro 12MP CCD camera (Delta Optical, Nowe Osiny, Poland).

### 4.5. Karyotyping Using Fluorescence In Situ Hybridization (FISH)

Karyotyping of *T. monococcum* accession PI 191383, rye cv. Hadron and einkorn-rye amphiploids was performed as described by Kwiatek et al. [40]. Three probes, namely, pTa-86, pTa-535, pTa713 [40,41] were used in FISH analysis. For chromosomal observation, we used DAPI (4′, 6-diamidino-2-phenylindole) as counterstain. Slides were analysed with the use of Axio Observer 7 (Carl Zeiss, Oberkochen, Germany) fluorescence microscope. Image processing was performed using ZEN Pro software (Carl Zeiss, Oberkochen, Germany). Each plant was evaluated by an analysis of chromosome sets of 10 cells. When capturing the signals, the slides were washed for the next ISH assay.

### 4.6. Genomic In Situ Hybridization (GISH)

Genomic in situ hybridization was performed according to Kwiatek et al. [42]. Chromosome sets of F_1_ hybrids and amphiploids (S_1_, S_2_ and S_3_) were analysed using genomic in situ hybridization. It was performed on mitotic chromosomes of root meristem cells collected from hybrid plants. Total genomic DNA was isolated using DNeasy Plant Maxi Kit 24 (Qiagen, Germany). DNA of *T. monococcum* subsp. *monococcum* cv. Laetissimum (2n = 2x = 14 chromosomes; A^m^A^m^; PI 191383; U.S. National Plant Germplasm System), a progenitor of the A-genome of wheat and triticale, was labelled by nick translation with Atto-488 dye (Atto-488NT kit; Jena Bioscience, Germany) for investigation of A^m^-genome chromosomes. Blocking DNA from rye (*S. cereale* L. cv. Imperial; PI 323382; U.S. National Plant Germplasm System) was sheared by boiling for 30–45 min and used at a ratio of 1:50 (probe:block).

### 4.7. Ptg Infection Test

In the experiments with Polish *Ptg* pathotypes mixture (Collection of Poznan University of Life Sciences), 10-day-old seedlings of following genotypes were examined: *T. monococcum* × *S. cereale* amphidiploids (S_4_ generation); *T. aestivum* “Thatcher”, (negative control, susceptible) and *T. aestivum* “Thatcher+ *Sr57/Lr34/Yr18/ Ltn1*” (GSTR 433; positive control, which possess a “slow rusting” gene, which provides a moderate resistance). Plants were grown in 4 × 8-cell horticultural plastic tray (30 cm × 60 cm) filled with universal soil in three repetitions (groups) and inoculated with 80 mg of uredinospores, using the spraying method [43]. All inoculated plants were kept in a humid glass chamber at 25 °C for incubation for the first 24 h and then shifted to a glass house chamber under 24 °C/20 °C (16 h day/8 h night) regime. Disease reaction was recorded for all accessions and different isolates at 8–10 days after the inoculation using 0–4 scale [28]. This rating system provides the following classifications: “immune” (rated as “0”), “very resistant” (rated as “;” or “0.5”), “resistant” (rated as “1”), “moderately resistant” (rated as “2”) “moderately resistant to moderately susceptible” (rated as “3”) and “susceptible” (rated as “4”) to steam rust. The means of stem rust expression scores for S_3_ amphiploids and control genotypes (between and within the groups) were compared using ANOVA calculations and Tukey’s HSD test and juxtaposed with PCR amplification products of *NL9F5* (*Sr35*) marker.

### 4.8. Evaluation of Yield Related Traits

The progeny (S_4_) of S_3_ generation plants obtained by self-pollination and parental accessions (*S. cereale* cv. ‘Dankowskie Hadron’ and *T. monococcum* PI 191383) were sown in five-row experimental plots (1 m^2^) in three repetitions at Agricultural Research Station of the PULS in Dłoń, Poland. Quantitative yield traits have been evaluated in 20 plants of each plot (3 genotypes × 20 plants × 3 repetitions). Following eight yield-shaping traits have been scored: plant height (PHt), stem length (StL), spike length (SpL), number of spikelets per spike (SplPS), number of grains per spike (GPS), grain length (GL), grain width (GW) and thousand-grain weight (TGW). Analyses of variance (ANOVA) for each trait data have been performed according to random model and compared between different accessions by the use of Tukey’s Honest Significant Difference (HSD) test at *p* = 0.05 and *p* = 0.01 significance levels (STATISTICA 13.1, StatSoft, Kraków, Poland).

## 5. Conclusions

In conclusion, it can be said that the scores for the yield-shaping parameters of the tetraploid triticale are comparable with the cultivated hexaploid triticale. Together with the efficient set of stem rust resistance genes, tetraploid triticale, presented here, has the potential to be exploited as a ‘bridge form’ to transfer the genetic-based resistance to commercial hexaploid triticale without lowering the biomass parameters, which meets the breeders’ expectations. Regarding the cytogenetic stability of the later generations of the given tetraploid triticale stock, it is also possible to consider this allopolyploid as an alternative industrial crop; however, this assumption needs future testing of technological characteristics of this synthetic cereal.

## Figures and Tables

**Figure 1 plants-12-00278-f001:**
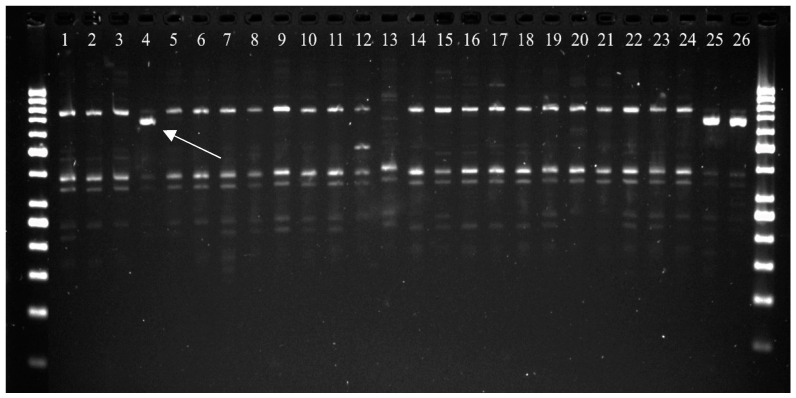
PCR products for *NL9F5* marker linked to *Sr35* stem rust resistance gene. 1–24—*T. monococcum* genotypes (Table 1). 25 and 26—positive controls (GSTR 526 and PI 428170). An arrow indicates a 719 bp product linked to resistance allele.

**Figure 2 plants-12-00278-f002:**
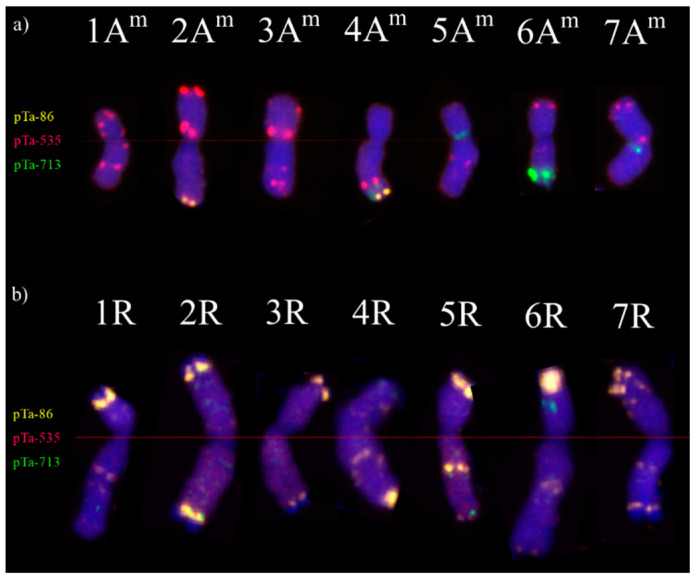
Karyotypes of (**a**) *T. monococcum* PI 191383 accession and (**b**) rye cv. Hadron. Yellow FISH signals are from probe pTa-86, red signals—pTa-535 and green signals are from probe pTa-713.

**Figure 3 plants-12-00278-f003:**
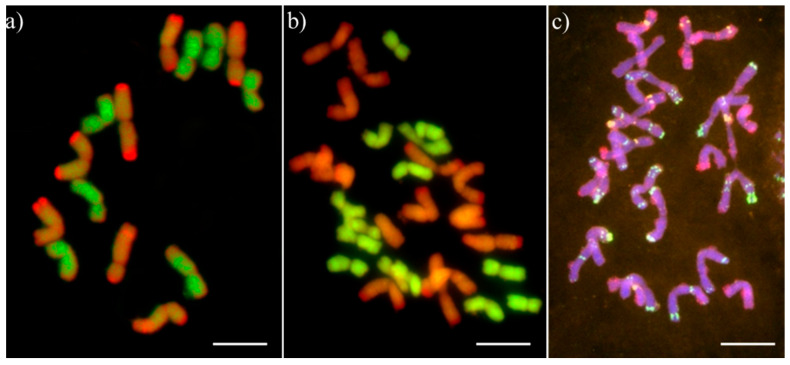
Chromosome sets analysed by genomic in situ hybridization (GISH) and fluorescent in situ hybridization (FISH) for (**a**) dihaploid plant (S_1_ generation) and (**b**) double haploid plant (S_2_ generation). A-genome chromosomes were labelled using digoxigenin-11-dUTP (green) and R-genome chromosomes were labelled with tetra-methyl 5dUTP-rhodamine (red). (**c**) FISH pattern of pTa86 (green) and pTa535 (red) probes on chromosomes of double haploid plant (S_2_ generation). Scale bar: 10 µm.

**Figure 4 plants-12-00278-f004:**
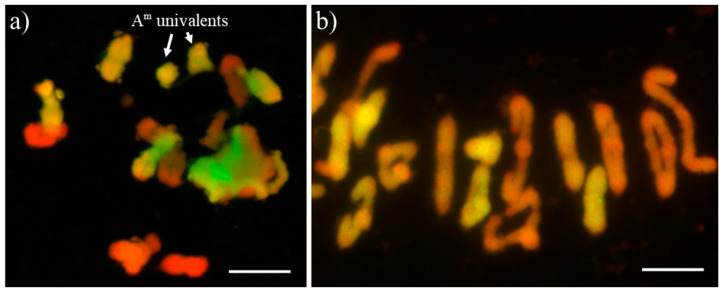
Chromosomes of tetraploid triticale at metaphase I of meiosis of pollen mother cells (PMCs) originated from (**a**) S_2_ generation plant; and (**b**) S_3_ generation plant. A-genome chromosomes were labelled using digoxigenin-11-dUTP (green) and R-genome chromosomes were labelled with tetra-methyl 5dUTP- rhodamine (red). Scale bars = 5 µm.

**Figure 5 plants-12-00278-f005:**
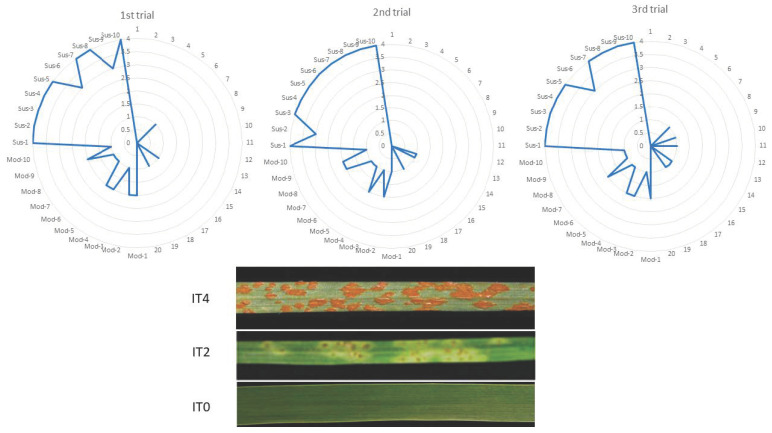
Infection scores collected from three experimental trials. Each trial consisted of 1–20 S_4_ and following control genotypes: *T. aestivum* “Thatcher”, (susceptible control, Sus1–Sus10) and *T. aestivum* Thatcher + *Sr57* (moderate resistant control, Mod1—Mod10). Infection types (ITs) were scored according to Stakman et al. (1962): “0”, “1”, “2”—resistant; “3” to “4”—susceptible.

**Figure 6 plants-12-00278-f006:**
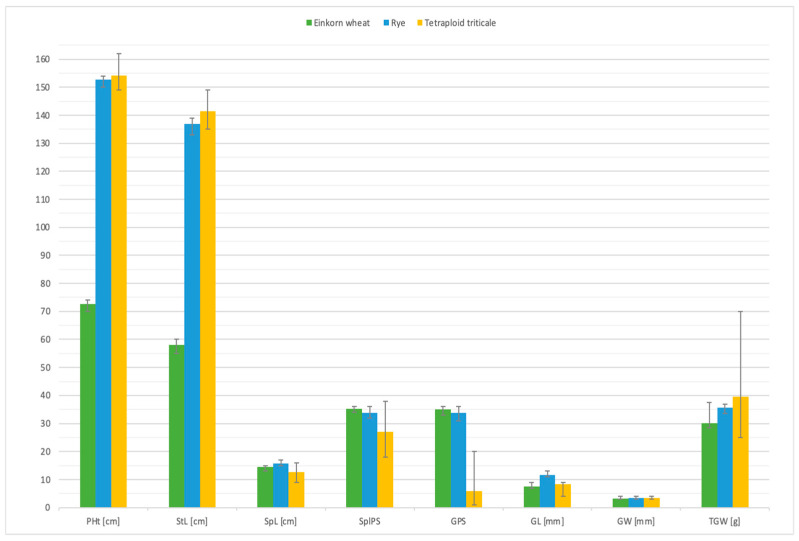
Graphical comparison of the mean and range scores for the following 8 yield-related traits investigated among *Triticum monococcum* (PI 191383), *Secale cereale* (Dankowskie Hadron) and tetraploid triticale (amphiploids—S_4_ generation): plant height (PHt), stem length (StL), spike length (SpL), number of spikelets per spike (SplPS), number of grains per spike (GPS), grain length (GL), grain width (GW) and thousand-grain weight (TGW).

**Figure 7 plants-12-00278-f007:**
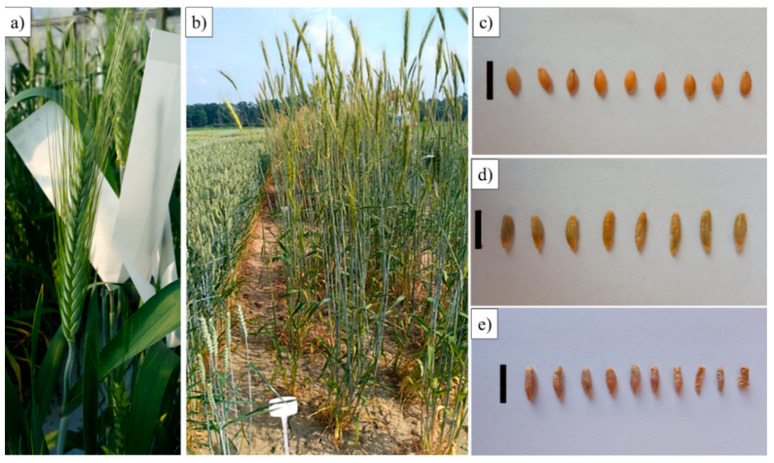
Spike (**a**) and whole plants (**b**) of S_4_ generation of tetraploid triticale. Seeds of: (**c**) *Triticum monococcum* (PI 191383), (**d**) *Secale cereale* (Dankowskie Hadron) and (**e**) tetraploid triticale (S_4_ generation). Scale bars: 1 cm.

**Table 1 plants-12-00278-t001:** *Triticum monococcum* and *Secale cereale* accessions screened with *Sr22* and *Sr35* markers.

No.	Accession Number	Name/Signature	Origin	PCR Products (Base Pairs) for *NL9F5* (*Sr35*) Marker
*Triticum monococcum* L. subsp. *monococcum*				
1.	PI 167627	3412	Turkey	800
2.	PI 190945	3962	Portugal	800
3.	PI 191381	Flavescens	Ethiopia	800
**4.**	**PI 191383**	**Laetissimum**	**Ethiopia**	**719**
5.	PI 277190	ATRI 613/59	no data	800
6.	PI 286068	no data	Poland	800
7.	PI 290508	V.J.388	Hungary	800
8.	PI 307984	K930	Morocco	800
9.	PI 326317	WIR18140	Azerbaijan	800
10.	PI 352480	Kelcyras	Albania	800
11.	PI 355515	6975.2	Asia Minor	800
12.	PI 355550	69Z5.40	Switzerland	800
13.	PI 427627	G2158	Turkey	800
14.	PI 427812	G2550	Iran	800
15.	PI 427927	G2824	Iraq	800
16.	PI 591871	H Tri 13605/87	Georgia	800
17.	PI 94740	287	Spain	800
18.	PI 94743	290	Russian Federation	800
19.	CLTR 13965	Metzger G68-32	U.S.A.	800
*Triticum monococcum* L. subsp. *aegilopoides* (Link) Thell.				
20.	PI 428011	G3224	Azerbaijan	800
21.	PI 428012	G3225	Armenia	800
22.	PI 554513	84TK154-02800	Russian Federation	800
23.	PI 614649	UKR-99-075	Ukraine	800
24.	PI 662221	GR05-052	Greece	800
*Secale cereale* L.				
25.	Piastowskie		Poland	0
26.	Poznańskie		Poland	0
27.	Antonińskie		Poland	0
28.	Hadron		Poland	0
29.	Granat		Poland	0
30.	Turkus		Poland	0

**Table 2 plants-12-00278-t002:** Cross-pollination efficiency (CE) and number of pollinated flowers and immature embryos derived for plant regeneration in in vitro conditions.

Einkorn (*Triticum monococcum*) PI 191383	Rye (*Secale cereale*) Cultivars (Pollen Donors)																	
	Granat			Hadron			Turkus			Antoninskie			Piastowskie			Poznańske		
*Spike Number*	*Number of Pollinated Florets*	*Number of Immature Embryos*	*CE*	*Number of Pollinated Florets*	*Number of Immature Embryos*	*CE*	*Number of Pollinated Florets*	*Number of Immature Embryos*	*CE*	*Number of Pollinated Florets*	*Number of Immature Embryos*	*CE*	*Number of Pollinated Florets*	*Number of Immature Embryos*	*CE*	*Number of Pollinated Florets*	*Number of Immature Embryos*	*CE*
1.	44	3	0.07	50	3	0.06	42	0	0.00	42	1	0.02	44	1	0.02	38	1	0.03
2.	42	1	0.02	44	4	0.09	38	2	0.05	40	0	0.00	42	0	0.00	36	4	0.11
3.	34	2	0.06	42	1	0.02	44	3	0.07	40	1	0.03	44	3	0.07	52	0	0.00
4.	42	4	0.10	44	5	0.11	38	3	0.08	38	3	0.08	42	3	0.07	50	1	0.02
5.	42	0	0.00	42	3	0.07	44	1	0.02	44	3	0.07	40	4	0.10	48	2	0.04
6.	38	4	0.11	44	1	0.02	38	2	0.05	44	1	0.02	40	4	0.10	52	4	0.08
7.	44	3	0.07	42	4	0.1	34	0	0.00	42	2	0.05	36	3	0.08	54	2	0.04
8.	38	3	0.08	40	0	0	34	3	0.09	40	1	0.03	52	0	0.00	52	0	0.00
9.	44	1	0.02	40	3	0.08	36	1	0.03	40	0	0.00	48	2	0.04	44	2	0.05
10.	38	0	0.00	36	3	0.08	34	2	0.06	36	3	0.08	46	2	0.04	42	1	0.02
11.	34	0	0.00	52	3	0.06	42	1	0.02	52	3	0.06	42	1	0.02	44	3	0.07
12.	34	3	0.09	50	0	0	44	2	0.05	50	4	0.08	44	0	0.00	42	3	0.07
13.	36	1	0.03	48	4	0.08	42	4	0.10	48	3	0.06	42	1	0.02	40	1	0.03
14.	34	2	0.06	52	3	0.06	44	1	0.02	46	3	0.07	44	3	0.07	40	1	0.03
15.	36	0	0.00	54	3	0.06	42	4	0.10	42	3	0.07	42	3	0.07	36	4	0.11
16.	36	2	0.06	52	2	0.04	40	0	0.00	46	0	0.00	40	3	0.08	42	3	0.07
17.	34	5	0.15	50	2	0.04	40	3	0.08	40	2	0.05	40	4	0.10	44	1	0.02
18.	36	2	0.06	52	1	0.02	36	3	0.08	46	1	0.02	36	1	0.03	44	2	0.05
19.	36	0	0.00	42	1	0.02	52	3	0.06	42	0	0.00	42	3	0.07	42	4	0.10
20.	44	3	0.07	40	1	0.03	50	0	0.00	42	2	0.05	44	0	0.00	40	3	0.08
*Summary:*	*766*	*39*	*0.05*	*916*	*47*	*0.05*	*814*	*38*	*0.05*	*860*	*36*	*0.04*	*850*	*41*	*0.05*	*882*	*42*	*0.05*

**Table 3 plants-12-00278-t003:** Subsequent self-pollinations of A^m^A^m^RR amphiploids.

No.	Crossing Combination	Number of Mature S_1_ Plants	Number of S_2_ Seeds	Number of Mature S_2_ Plants	Number of S_3_ Seeds	Number of Mature S_3_ Plants
1.	PI 191383 × Granat	19	7	0	0	0
2.	PI 191383 × Hadron	26	5	2	79	68
3.	PI 191383 × Turkus	24	8	0	0	0
4.	PI 191383 × Antoninskie	13	5	0	0	0
5.	PI 191383 × Piastowskie	31	4	0	0	0
6.	PI 191383 × Poznańskie	28	9	0	0	0
**Summary**		**141**	**38**	**2**	**79**	**68**

**Table 4 plants-12-00278-t004:** Chromosome configurations during meiosis of Pollen Mother Cells (PMCs) at metaphase I analysed for S_2_ and S_3_ generation plants.

Generation	Plant No.	Number of Analysed PMCs	Mean Number of Bivalents		Mean Number of Univalents		Mean Number of Multivalents	
			*T. monococcum*	*S. cereale*	*T. monococcum*	*S. cereale*	*T. monococcum*	*S. cereale*
S_2_	S2_1	10	6.8	4.8	0.4	0	0	1.2
	S2_2	10	6.6	4.9	0.8	0	0	1.3
**summary**	**2**	**20**	**6.7**	**4.85**	**0.6**	**0**	**0**	**1.25**
S_3_	S3_1	10	6.9	6.8	0.2	0	0	0.1
	S3_2	10	6.8	6.8	0.4	0	0	0.1
	S3_3	10	7	7	0	0	0	0
	S3_4	10	7	7	0	0	0	0
	S3_5	10	6.9	6.8	0.2	0	0	0.1
	S3_6	10	7	7	0	0	0	0
	S3_7	10	7	7	0	0	0	0
	S3_8	10	6.8	6.8	0.4	0	0	0.1
	S3_9	10	7	7	0	0	0	0
	S3_10	10	7	7	0	0	0	0
**summary**	**10**	**100**	**6.94**	**6.92**	**0.12**	**0**	**0**	**0.04**

**Table 5 plants-12-00278-t005:** Infection scores compared with PCR amplification products *NL9F5* (*Sr35*) marker. *, indicates an allele which confers resistance.

Genotypes	Plants	Infection Scores			PCR Products (Base Pairs) *NL9F5 (Sr35)*
		*Group 1*	*Group 2*	*Group 3*	
A^m^A^m^RR amphiloids (S_3_ generation)	1	0	0.5	0	719 *
	2	0	0.5	0	719 *
	3	0	0	0.5	719 *
	4	0.5	0	0.5	719 *
	5	0.5	0.5	0.5	719 *
	6	1	0	1	719 *
	7	0.5	0	0.5	719 *
	8	0	0.5	0	719 *
	9	0.5	0.5	1	719 *
	10	0	0.5	0.5	719 *
	11	0	0.5	1	719 *
	12	0.5	0.5	0.5	719 *
	13	0	1	0.5	719 *
	14	0.5	1	0.5	719 *
	15	1	0.5	1	719 *
	16	0.5	0.5	1	719 *
	17	0.5	0	1	719 *
	18	1	1	0.5	719 *
	19	0.5	0	0	719 *
	20	0	0.5	0.5	719 *
*T. aestivum* Thatcher + *Sr57*	Mod-1	2	1	2	800
	Mod-2	2	2	1	800
	Mod-3	1	1	2	800
	Mod-4	2	2	2	800
	Mod-5	2	1	1	800
	Mod-6	1	1	1	800
	Mod-7	1	1	2	800
	Mod-8	1	2	1	800
	Mod-9	2	2	1	800
	Mod-10	1	1	1	800
*T. aestivum* “Thatcher”	Sus-1	4	4	4	800
	Sus-2	4	3	4	800
	Sus-3	4	4	4	800
	Sus-4	4	4	4	800
	Sus-5	4	4	4	800
	Sus-6	3	4	3	800
	Sus-7	4	4	4	800
	Sus-8	4	4	4	800
	Sus-9	3	4	4	800
	Sus-10	4	4	4	800

**Table 6 plants-12-00278-t006:** Statistics, analysis of variance followed by Tukey’s Honest Significant Difference test (HSD) of 8 yield-related traits investigated among 60 plants of *Triticum monococcum* (PI 191383), Secale cereale (Dankowskie Hadron) and amphiploids (S4 generation): plant height (PHt), stem length (StL), spike length (SpL), number of spikelets per spike (SplPS), number of grains per spike (GPS), grain length (GL), grain width (GW) and thousand-grain weight (TGW).

Genotypes	Summary Values	PHt [cm]	StL [cm]	SpL [cm]	SplPS	GPS	GL [mm]	GW [mm]	TGW [g]
Einkorn wheat *Triticum monococcum* A^m^API 191383	Mean	72.5333	58.1	14.4333	35.2667	35.1	7.6167	3.2333	30.2
	Range	70–74	55–60	13–15	34–36	33–36	7–9	3–4	28.4–37.6
	Variance	1.2023	2.0237	0.4192	0.9446	1.0407	0.4438	0.1819	5.7464
	Standard deviation	1.0965	1.4226	0.6475	0.9719	1.0201	0.6662	0.4265	2.3972
	Standard error	0.1416	0.1837	0.0836	0.1255	0.1317	0.086	0.0551	0.3095
Rye *Secale cereale* RR Dankowskie Hadron	Mean	152.7833	136.95	15.8333	33.8333	33.7167	11.5333	3.3667	35.7533
	Range	150–154	133–139	15–17	32–36	31–36	11–13	3–4	33.6–36.9
	Variance	0.9862	2.2517	0.75514	1.9379	2.0031	0.4565	0.2362	0.732
	Standard deviation	0.9931	1.5006	0.8668	1.9321	1.4153	0.6756	0.486	0.8556
	Standard error	0.4282	0.1937	0.1119	0.1797	0.1827	0.0872	0.0627	0.1105
Amphiploids A^m^A^m^RR S_4_ plants	Mean	154.1833	141.45	12.75	26.9667	5.8833	8.2833	3.3167	39.4967
	Range	149–162	135–149	9–16	18–38	1–20	4–9	3–4	25.1–70.2
	Variance	12.254	15.8449	3.2754	19.1175	20.4438	1.2573	0.2201	54.219
	Standard deviation	3.5006	3.9806	1.8098	4.3724	4.5215	1.1213	0.4691	7.3634
	Standard error	0.4519	0.5139	0.2336	0.5645	0.5837	0.1448	0.0606	0.9506
**Analysis of variance (ANOVA)—treatment between groups**									
F–value		27,229.76	19,658.9	96.49	161.04	2082.24	366.34	1.28	64.89
*p*–value		<0.0001	<0.0001	<0.0001	<0.0001	<0.0001	<0.0001	0.285598	<0.0001
**Tukey’s Highest Significant Difference (HSD) between any two sample means at designated level**									
A^m^A^m^ vs. RR		*p* < 0.01	*p* < 0.01	*p* < 0.01	*p* < 0.05	*p* < 0.05	*p* < 0.01	n/a	*p* < 0.01
A^m^A^m^ vs. A^m^A^m^RR		*p* < 0.01	*p* < 0.01	*p* < 0.01	*p* < 0.01	*p* < 0.01	*p* < 0.01	n/a	*p* < 0.01
RR vs. A^m^A^m^RR		*p* < 0.01	*p* < 0.01	*p* < 0.01	*p* < 0.01	*p* < 0.01	*p* < 0.01	n/a	*p* < 0.01
**Significance level**	HSD [0.05]	0.95	1.12	0.53	1.17	1.21	0.37	n/a	1.94
	HSD [0.01]	1.18	1.39	0.66	1.46	1.51	0.46	n/a	2.42

## Data Availability

The data presented in this study are available on request from the corresponding author.

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
