# Peer review of "Novel Tetraploid Triticale (Einkorn Wheat × Rye)—A Source of Stem Rust Resistance"

_plants, 2023, doi:10.3390/plants12020278_

Round 1

Reviewer 1 Report

This manuscript seems reports on interesting research, which seems to have been conducted properly. However, substantial editing is required to correct the English.  Scientific review should be deferred until after the  text has been revised.

Author Response

We have removed the grammar errors, and rewrite several sentences to be more clear for the readers.

Considering that we have used wrong or old markers for Sr21 and Sr22 we have decided to remove those results. We decided to present the results for the Sr35 marker, taking into consideration that it is based on the cloned gene.

We have added an additional paragraph, which describes Pgs in the context of stem rust resistance in triticale.

Reviewer 2 Report

This paper describing AARR novel triticales has a lot of potential and will receive a lot of interest from readers. The strength of the paper is in the description of the genetics of the AARR triticales and supporting GISH and agronomic data. The stem rust component of the paper needs to be substantially improved before I would recommend accepting this manuscript. Here are a few items that need to be addressed:

1. The English writing is poor. Please have the manuscript reviewed appropriately.

2. Use Pgt, not Ptg.

3. The isolates of Pgt need to be described. Inoculating with a bulk inoculum is not ideal. The isolates need to be referenced with an isolate name and where they are stored (this is important for repeatability). The avirulence/virulence of the isolate to Sr genes should be described, especially for the relevant monococcum Sr genes, Sr21, Sr22, and Sr35 (great if you also have data for Sr22b and Sr60, but  not necessary).

4. There is a Sr22b gene described from monococcum in addition to Sr22 which is now called Sr22a.

5. The SSR marker linked with Sr22 cannot be used to predict this gene in monococcum germplasm. This marker is only useful in wheat germplasm. Sr22a and Sr22b have both been cloned and there are better markers the authors should use. The marker used for Sr35 is based on the cloned gene and is appropriate.

6. The PI 191383 line does not have Sr22. The reference you cite screened the line with appropriate race to be able to postulate Sr22 and did not postulate Sr22. Assigning Sr22 to Pi 191383 based on the SSR marker alone is not sufficient.

7. The introduction section needs to describe Pgt vs the rye stem rust pathogen Pgs and the role of Pgt vs. Pgs on triticale in Europe. What is known about stem rust on triticale and rye in Europe. Is it a problem?

8. Figure 6 needs error bars.

9. There are markers linked to Sr21 available (this has been cloned), the authors are welcome to track Sr21 based on good markers in their germplasm - this is notable since PI 191383 was postulated to possess Sr21, not Sr22.

Author Response

Respond to reviewers

We would like to thank the reviewers for their thoughtful comments and efforts towards improving our manuscript. Your detailed reviews showed that the manuscript needs to be improved significantly. In view of the number and diversity of your comments, and looking at the multitude and the range of questions regarding our manuscript, I have listed our answers below:

  1. The English writing is poor. Please have the manuscript reviewed appropriately.

We have removed the grammar errors, and rewrite several sentences to be more clear for the readers.

  1. Use Pgt, not Ptg.

We have change Ptg shortcut into the correct one – Pgt.

  1. The isolates of Pgt need to be described. Inoculating with a bulk inoculum is not ideal. The isolates need to be referenced with an isolate name and where they are stored (this is important for repeatability). The avirulence/virulence of the isolate to Sr genes should be described, especially for the relevant monococcum Sr genes, Sr21, Sr22, and Sr35 (great if you also have data for Sr22b and Sr60, but not necessary).

We agree that the use of local races of Pgt is not the perfect testing. We wanted to check if there is any significant difference between the infection responses of amphidiploids (S4 generation); T. aestivum “Thatcher”, (negative control, susceptible) and T. aestivum “Thatcher+ Sr57/Lr34/Yr18/ Ltn1” (GSTR 433; positive control, which possess a “slow rusting” gene, which provides a moderate resistance). This approach provedd, that the tetraploid triticale is showing a high level of resistance against stem rust. Our next goal is to conduct a set of specific virulence test using relevant isolates

  1. There is a Sr22b gene described from monococcum in addition to Sr22 which is now called Sr22a.
  2. The SSR marker linked with Sr22 cannot be used to predict this gene in monococcum germplasm. This marker is only useful in wheat germplasm. Sr22a and Sr22b have both been cloned and there are better markers the authors should use. The marker used for Sr35 is based on the cloned gene and is appropriate.
  3. The PI 191383 line does not have Sr22. The reference you cite screened the line with appropriate race to be able to postulate Sr22 and did not postulate Sr22. Assigning Sr22 to Pi 191383 based on the SSR marker alone is not sufficient.
  4. There are markers linked to Sr21 available (this has been cloned), the authors are welcome to track Sr21 based on good markers in their germplasm - this is notable since PI 191383 was postulated to possess Sr21, not Sr22.

Considering that we have used wrong or old markers for Sr21 and Sr22 we have

decided to remove those results. Many thanks for you suggestions and information.

  1. The introduction section needs to describe Pgt vs the rye stem rust pathogen Pgs and the role of Pgt vs. Pgs on triticale in Europe. What is known about stem rust on triticale and rye in Europe. Is it a problem?

We have added an additional paragraph, which describes Pgs in the context of stem rust resistance in triticale.

  1. Figure 6 needs error bars.

We have added bars into the graph (fig. 6)

Round 2

Reviewer 1 Report

My main concern was about the previous version was the need to substantially improve English expression. Because of this, I was not able to review the scientific content of the manuscript. Major improvements have now been made to the writing. The revised manuscript is reasonably easy to understand and it looks scientifically sound.

It is a bit disappointing to see that the response to some comments from the other reviewer was to remove parts of the manuscript rather than to obtain and apply more appropriate markers.

I note that the manuscript indicates that data are available upon request. This is good, but what about the materials?

I trust that the publisher's editorial staff will go through the manuscript carefully to correct remaining grammatical errors and rephrase awkwardly worded passages.

Author Response

Many thanks for your second review. We appreciate your time and effort.

The plant material are the subject of the implementation agreement between University ant the breeding company. The materials can be shared after MTA signing.

All the best,

Michał Kwiatek